# Antimicrobial Susceptibility in Respiratory Pathogens and Farm and Animal Variables in Weaned California Dairy Heifers: Logistic Regression and Bayesian Network Analyses

**DOI:** 10.3390/antibiotics13010050

**Published:** 2024-01-04

**Authors:** Brittany L. Morgan Bustamante, Munashe Chigerwe, Beatriz Martínez-López, Sharif S. Aly, Gary McArthur, Wagdy R. ElAshmawy, Heather Fritz, Deniece R. Williams, John Wenz, Sarah Depenbrock

**Affiliations:** 1Public Health Sciences, School of Medicine, University of California, Davis, Davis, CA 95616, USA; 2Center for Animal Disease Modeling and Surveillance, Department of Veterinary Medicine and Epidemiology, School of Veterinary Medicine, University of California, Davis, Davis, CA 95616, USA; 3Department of Veterinary Medicine and Epidemiology, School of Veterinary Medicine, University of California, Davis, Davis, CA 95616, USA; 4Veterinary Medicine Teaching and Research Center, School of Veterinary Medicine, University of California, Davis, Tulare, CA 93274, USA; 5Department of Population Health and Reproduction, School of Veterinary Medicine, University of California, Davis, Davis, CA 95616, USA; 6Swinging Udders Veterinarian Services, Galt, CA 95632, USA; 7Department of Internal Medicine and Infectious Diseases, Faculty of Veterinary Medicine, Cairo University, Giza 12613, Egypt; 8California Animal Health and Food Safety Laboratory System, School of Veterinary Medicine, University of California, Davis, Davis, CA 95616, USA; 9Field Disease Investigation Unit, College of Veterinary Medicine, Washington State University, Pullman, WA 99163, USA

**Keywords:** Bayesian network analysis, *Pasteurella multocida*, *Mannheimia haemolytica*, *Histophilus somni*, antibiotic, antimicrobial drug, antimicrobial resistance, dairy management practices, bovine respiratory disease

## Abstract

Weaned dairy heifers are a relatively understudied production group. Bovine respiratory disease (BRD) is the most common cause of antimicrobial drug (AMD) use, morbidity, and mortality in this production group. The study of antimicrobial resistance (AMR) is complicated because many variables that may affect AMR are related. This study generates hypotheses regarding the farm- and animal-level variables (e.g., vaccination, lane cleaning, and AMD use practices) that may be associated with AMR in respiratory isolates from weaned dairy heifers. A cross-sectional study was performed using survey data and respiratory isolates (*Pasteurella multocida*, *Mannheimia haemolytica*, and *Histophilus somni*) collected from 341 weaned dairy heifers on six farms in California. Logistic regression and Bayesian network analyses were used to evaluate the associations between farm- and animal-level variables with minimum inhibitory concentration (MIC) classification of respiratory isolates against 11 AMDs. Farm-level variables associated with MIC classification of respiratory isolates included the number of source farms of a calf-rearing facility, whether the farm practiced onsite milking, the use of lagoon water for flush lane cleaning, and respiratory and pinkeye vaccination practices. Animal-level variables associated with a MIC classification included whether the calf was BRD-score-positive and time since the last phenicol treatment.

## 1. Introduction

Bovine respiratory disease (BRD) is among the most important diseases of cattle across multiple management systems and is among the most common reasons for antimicrobial drug (AMD) use in cattle [1,2,3]. Weaned dairy heifers are an economically important, yet relatively understudied animal production group, and BRD is the most common cause of AMD use, morbidity, and mortality in this production group [1]. Antimicrobial resistance (AMR) is highly prevalent in this production group [4], and has increased among BRD pathogens in other management systems for many drugs used to treat or control BRD [5]. Antimicrobial use drives AMR [6]; however, AMD use alone does not explain all AMR outcomes; other management factors may influence AMR [7]. The study of factors that drive AMR is complex because multiple genetic elements that confer different mechanisms of AMR can be shared between bacteria, and bacteria can be transmitted easily between animals, humans, and the environment. The differential selective pressures that can be exerted on the host, pathogen, and environment create a complex system in which AMR can develop.

A primary goal of most epidemiological studies is identifying risk factors for disease outcomes; regression analysis is the most commonly used approach in this endeavor [8]. However, when the condition under study is complex, researchers often deal with correlations among their predictor variables and potentially multiple, correlated outcome variables. A multivariate modeling method, such as Bayesian network analysis (BNA), has the potential to consider variable interdependence and demonstrate novel epidemiological insights in addition to what may be identified by classical regression approaches [9,10,11]. Bayesian networks are probabilistic graphical models representing a set of random variables and their conditional dependencies. These graphical models are often depicted as directed acyclic graphs (DAGs) composed of nodes (random variables in the study) and edges (indicating statistical dependency between variables). Furthermore, Bayesian networks may be particularly well-suited to studies exploring AMR because bacteria are rarely resistant or susceptible to a single AMD; instead, complex patterns of resistance to different AMDs simultaneously are commonly reported [4,12,13]. 

Previous studies have outlined ways to identify AMR profiles using BNA, with the recommendation that these profiles of resistance be included in potential risk factor analyses [10,14,15]. This study uses both logistic regression and BNA to describe associations between farm- and animal-level variables and AMR in respiratory bacterial isolates from a population of California weaned dairy heifers. 

## 2. Results

Samples and data from a total of 341 calves were included in the final analysis; 19 calf samples and associated data were removed from the analysis due to missing data, label errors, or other data entry errors. A total of 145, 119, and 97 *P. multocida, M. haemolytica,* and *H. somni* (respectively) isolates (n = 361) were cultured from these 341 calves. The percentage of all isolates classified as not susceptible were as follows: tilmicosin (55.1%), tildipirosin (52.1%), tetracycline (45.4%), gamithromycin (41.0%), enrofloxacin (48.5%), danofloxacin (47.9%), florfenicol (34.9%), spectinomycin (33.8%), tulathromycin (16.6%), penicillin (12.5%) and ceftiofur (0%). The majority of *P. multocida* isolates were classified as not susceptible to tetracycline (100%), gamithromycin (69.4%), tildipirosin (73.6%), tilmicosin (76.0%), florfenicol (59.5%), enrofloxacin (57.9%) and danofloxacin (57.9%). Most *M. haemolytica* isolates were classified as not susceptible to tildipirosin (61.8%), danofloxacin (68.2%), enrofloxacin (69.1%), tetracycline (86.4%), and tilmicosin (61.8%). *H. somni* isolates were most frequently classified as not susceptible to spectinomycin and tetracycline (70.1% and 62.9%, respectively). Details of MIC testing and breakpoint analysis in this study population have been previously published [4].

Of the six farms enrolled, three were multisource calf-rearing facilities with a range of eight to approximately 45 source farms, two facilities raised their calves at the source dairy, and one facility raised their weaned heifers back at the source dairy after calves returned from a multisource calf-rearing facility which was used for hutched/milk-fed calves. Four of the six farms had onsite milking. Vaccines administered to calves <6 months of age at each facility included: commercial *Salmonella* (*Salmonella* Dublin or *Salmonella* Newport), commercial respiratory modified live viral combination (bovine viral diarrhea virus, bovine respiratory syncytial virus, and infectious bovine rhinotracheitis, parainfluenza-3), a commercial clostridial combination (*Clostridium chauvoei, septicum, novyi, sordellii,* and *perfringens* types C and D), *Brucella abortus*, inconsistent combinations of commercial and autogenous vaccinations for pinkeye (bovine keratoconjunctivitis, including *Moraxella bovis* +/− *bovoculi*) for farms that vaccinated for this disease. Other biologicals included an oral antibody product (Corona virus/*E. coli*). Colostrum was heat-treated on five of six farms. All farms fed hospital milk, although specific practices for volume, frequency, and nutritional components for milk-feeding varied widely by facility and calf age. Hospital milk, also known as waste milk or milk that is non-saleable due to disease or treatment in the cow, was pasteurized on five farms. Bull calves were not comingled with heifer calves in three facilities. Group medication varied by farm and included: no medication or intermittent use of sulfamethoxazole or neomycin by group. All grain fed to calves on all study farms was medicated with either an ionophore (monensin; a non-medically important AMD), coccidiostat (amprolium), a combination of monensin and amprolium, or a chlortetracycline/ sulfamethazine type A medicated feed. Water was reported to be historically medicated in animal groups for five of six farms with tetracycline or sulfadimethoxine on an as-needed basis, although recent use of this practice in the 6 months preceding sampling was not reported by management. Feed lane cleaning methods for weaned heifer group pens consisted of scraping (2/6 farms), flushing with lagoon water (2/6 farms) or clean water (1/6 farms), or a combination of scraping and flushing with lagoon water (1/6 farms). 

Most (64.0%) of the isolates included in the study were from a calf-rearing facility that sourced calves from multiple locations. Most isolates were from calves that resided on a farm with onsite milking (70.9%), where *Salmonella* and pinkeye vaccines were administered (46.2% and 69.0%, respectively), and where hospital milk was pasteurized (82.0%). Almost half of the isolates were from heifers on farms where bull calves were co-mingled (49.0%), and the majority were fed grain that was medicated (13.0% of isolates with chlortetracycline/sulfamethazine type A medicated feed, 51.0% of isolates with amprolium, and 87.0% of isolates with monensin). Water was known to be medicated for groups at some time during heifer-rearing with tetracycline or sulfadimethoxine for 83.1% and 65.1% of isolates, respectively. Feed lane cleaning methods were mostly scrape or lagoon flush, with only 19.1% of isolates coming from calves residing on a farm where clean water was used. Dust management plans were in place on farms where 82% of isolates were obtained. There were 13,229, 14,466, 53,418, and 6087 calves born or arriving on each of the four farms with reliable AMD treatment data over a year, from which the AMD treatment frequency for all calves from birth to six months of age was obtained (Figure 1). A summary of the farm-level variables included in the analysis is presented in Table 1.

Reliable AMD treatment history information was available for 263 calves, which were included in the animal-level analysis. The individual AMD treatment history of sampled calves was variable; 11.0% of isolates came from calves that had been treated at least once prior to sampling with a cephalosporin, 29.7% with a macrolide, 58.2% with a phenicol, 17.5% with a tetracycline, 22.8% with a penicillin, 7.6% with a sulfonamide, and 32.3% with a fluoroquinolone. The interval between the last drug treatment and sampling was variable due to the nature of naturally occurring disease events. Approximately half of the isolates came from calves that were BRD-score-positive at the time of sampling (52.1%). Summary statistics for the factors included in the animal-level analysis, stratified by respiratory pathogens, can be found in Table 2.

### 2.1. Logistic Regression

The logistic regression models are summarized in Table 3.

#### 2.1.1. *P. multocida*: Farm-Level Variables

Vaccination status (no vaccination) with commercial modified live injectable respiratory viral combination vaccine 1 (IRV1) (*p* < 0.0001) or commercial modified live injectable respiratory viral combination vaccine 3 (IRV3) (*p* < 0.0001) was associated with *P*. *multocida* isolates being not susceptible to fluoroquinolones, when compared to vaccinating with IRV2. The percentage of *P*. *multocida* isolates predicted to be resistant to fluoroquinolones when not vaccinating with IRV1 and IRV3 was 58.7% and 57.8%, respectively. Not vaccinating with IRV3 was positively associated with an MIC classification of not susceptible for *P*. *multocida* isolates to florfenicol (*p* < 0.0001) and macrolides (*p* < 0.0001). Feeding lane cleaning (*p* < 0.0001) was associated with MIC classification for susceptibility of *P*. *multocida* to spectinomycin. The predicted percentage of *P*. *multocida* isolates resistant to spectinomycin was 4.6% when feeding lane cleaning was performed by scraping and clean water flushing compared to 56.4% when feeding lane cleaning was performed by scraping and lagoon water flushing. Analysis was not performed for susceptibility patterns for cephalosporins, penicillin, and tetracyclines because all isolates were susceptible, only a single isolate was not susceptible, and no isolates were susceptible, respectively. 

#### 2.1.2. *P. multocida*: Animal-Level Variables

The absence of previous treatment with tetracycline (*p* = 0.01) and sulfonamides (*p* = 0.01) was associated with susceptibility of *P*. *multocida* to fluoroquinolones. The predicted percentage of *P*. *multocida* isolates not susceptible to fluoroquinolones after no previous treatment with tetracyclines and sulfonamides was 29% and 7.4%, respectively. Treatment with tetracyclines (*p* < 0.0001) and the season during which the sample was collected (*p* = 0.03) were associated with MIC classification of *P*. *multocida* to florfenicol. The predicted percentage of *P*. *multocida* isolates not susceptible to florfenicol when animals were previously treated with tetracycline compared to no treatment was 80.6% and 12.4%, respectively. This result should be interpreted cautiously because all *P*. *multocida* isolates were classified as not susceptible to tetracycline. 

The predicted percentage of *P*. *multocida* isolates not susceptible to florfenicol in the hot and cool seasons was 80.6% and 61%, respectively. A positive BRD score (*p* = 0.02) and treatment with sulfonamides (*p* < 0.0001) was negatively associated with MIC classification of *P*. *multocida* to macrolides. The predicted percentage of *P*. *multocida* isolates not susceptible to macrolides for calves with a positive and negative BRD score was 85.7% and 96.7%, respectively. The predicted percentage of *P*. *multocida* isolates not susceptible to macrolides after previous treatment with sulfonamides was 96.7%, compared to 59.5% for no treatment. Logistic regression was not performed for susceptibility patterns for cephalosporins, penicillin, and tetracyclines because all isolates were classified as susceptible, only two isolates were not susceptible, and all isolates were classified as not susceptible, respectively.

#### 2.1.3. *M. haemolytica*: Farm-Level Variables

Feeding lane cleaning (*p* < 0.0001) was associated with MIC classification for *M*. *haemolytica* isolates to florfenicol (*p* < 0.0001) and penicillin (*p* < 0.0001). The predicted percentage of *M. haemolytica* isolates not susceptible to florfenicol was 19.8% when feeding lane cleaning was performed by scraping and clean water flushing, compared to 64.9% when feeding lane cleaning was performed by scraping and lagoon water flushing. The predicted percentage of *M*. *haemolytica* isolates not susceptible to penicillin was 16.7% when feeding lane cleaning was performed by scraping and clean water flushing compared to 71.2% when feeding lane cleaning was performed by scraping and lagoon water flushing. Comingling heifer and bull calves was associated with MIC classification for *M*. *haemolytica* isolates to spectinomycin (*p* < 0.0004); the predicted percentage of *M*. *haemolytica* isolates not susceptible to spectinomycin was 32.7% when heifer and bull calves were comingled compared to 3% when not comingled. Not vaccinating with IRV1 (*p* = 0.007) was positively associated with MIC classification for *M*. *haemolytica* to tetracyclines. The predicted percentage of *M*. *haemolytica* isolates not susceptible to tetracyclines was 96.5% when not vaccinated with IRV1 compared to 77% when vaccinated with IRV1. 

Comingling heifer and bull calves (*p* = 0.001) and group medication in water (*p* = 0.02) were associated with MIC classification of *M*. *haemolytica* isolates to macrolides; the predicted percentage of *M*. *haemolytica* isolates not susceptible to macrolides was 57.7% when heifer and bull calves were not comingled compared to 85.2% when comingled. The predicted percentage of *M*. *haemolytica* isolates not susceptible to macrolides when water was group-medicated with tetracycline and sulfonamides was 90.7% compared to tetracycline and amprolium or no medication (76.7%). There was no association between farm-level variables and MIC classification for *M*. *haemolytica* isolates to fluoroquinolones (*p* = 0.99). Logistic regression analysis was not performed for susceptibility patterns for cephalosporins because all isolates were classified as susceptible.

#### 2.1.4. *M. haemolytica*: Animal-Level Variables

The absence of previous treatment with tetracycline (*p* = 0.0003) was associated with MIC classification of *M. haemolytica* to fluoroquinolones; the predicted percentage of *M. haemolytica* isolates not susceptible to fluoroquinolones after previous treatment and no treatment with tetracyclines was 78.1% and 7.1%, respectively. The season during which the sample was collected (*p* = 0.01), and previous treatment with tetracycline (*p* = 0.01) or fluoroquinolones (*p* = 0.007) were associated with MIC classification of *M. haemolytica* to florfenicol; the predicted percentage of *M*. *haemolytica* isolates not susceptible to florfenicol in the hot and cool seasons was 32.3% and 12.6%, respectively. The absence of previous treatment with tetracyclines was associated with a classification of *M. haemolytica* of susceptible to florfenicol; the predicted percentage of *M. haemolytica* isolates classified as susceptible to florfenicol after previous treatment with tetracyclines compared to no treatment with tetracyclines was 44.3% and 12.6%, respectively. For isolates from animals not previously treated with fluoroquinolones, the predicted percentage of non-susceptibility to fluoroquinolones was 12.6% compared to 35.4% for those from animals treated. The absence of treatment with tetracycline (*p* = 0.001) was associated with MIC classification of *M. haemolytica* to penicillin; the predicted percentage of *M. haemolytica* isolates classified as not susceptible to penicillin after previous treatment and no treatment with tetracyclines was 92.3% and 28.1%, respectively.

The season during which the sample was collected (*p* = 0.04), and previous treatment with macrolides (*p* = 0.006) were associated with MIC classification of *M. haemolytica* to macrolides; the predicted percentage of *M. haemolytica* isolates not susceptible to macrolides in the hot and cool seasons was 44.3% and 65%, respectively. The predicted percentage of *M*. *haemolytica* isolates not susceptible to macrolides after previous treatment and no treatment with macrolides was 92% and 65%, respectively. There was no association between MIC classification of *M. haemolytica* to tetracyclines and animal-level variables. Logistic regression was not performed for susceptibility patterns for cephalosporins because all isolates were susceptible. 

#### 2.1.5. *H. somni*: Farm-Level Variables

Pasteurization of colostrum (*p* < 0.0001) and the absence of vaccination with IRV1 (*p* = 0.002) were associated with MIC classification for *H*. *somni* isolates to tetracyclines. The predicted percentage of *H*. *somni* isolates not susceptible to tetracyclines when pasteurized colostrum was fed and vaccination with IRV1 was not performed was 46.3%, compared to 61.2% when unpasteurized colostrum was fed, and vaccination with IRV1 was performed. However, the predicted percentage of *H*. *somni* isolates not susceptible to tetracyclines was 19.9% when pasteurized colostrum was fed, and vaccination with IRV1 was performed. Furthermore, the predicted percentage of *H*. *somni* isolates not susceptible to tetracyclines was 84.5% when unpasteurized colostrum was fed, and vaccination with IRV1 was not performed.

Logistic regression analysis was not performed for cephalosporins, fluoroquinolones, and florfenicol because all isolates were susceptible. Logistic regression analysis was not performed for susceptibility classifications for penicillin because only three isolates were not susceptible. The association between colostrum pasteurization, vaccination with IRV1, and MIC classification for *H*. *somni* isolates susceptibility to tetracyclines was farm d-pendent. There was no association between any farm variables and MIC classification for *H. somni* isolates to macrolides.

#### 2.1.6. *H. somni*: Animal-Level Variables

There was no association between animal-level variables and the MIC classification of *H. somni* to tetracyclines or macrolides. Logistic regression was not performed for MIC classification for cephalosporins, fluoroquinolones, florfenicol, and penicillin because all isolates were susceptible.

### 2.2. Bayesian Network Analysis

#### 2.2.1. *P. multocida*: Farm-Level Variables

Farm-level practices linked to MIC classification include onsite milking and feed lane cleaning methods in weaned pens. The absence of onsite milking was associated with the classification of not susceptible for *P. multocida* to tildipirosin. The probability of tildipirosin non-susceptibility, if onsite milking occurred, was 69%, compared to 96% if the farm did not practice onsite milking. This association was further influenced by the feed lane cleaning method. The probability of tildipirosin non-susceptibility, given onsite milking and pens cleaned by scraping, was 83%. For a farm using a clean water source for pen cleaning, the probability of tildipirosin non-susceptibility, with onsite milking, was only 17%. Without onsite milking, the probability of tildipirosin non-susceptibility given clean water pen cleaning was 71%, lower than that of tildipirosin non-susceptibility without onsite milking for farms using scraping as a pen-cleaning method (*p* = 97%). 

Classification of non-susceptibility to tildipirosin is also associated with non-susceptibility to gamithromycin, as demonstrated in Figure 2. The probability that an isolate was not susceptible to gamithromycin, if it was not susceptible to tildipirosin, was 95%, compared to just 1% if the isolate was susceptible to tildipirosin. The categorization of not susceptible for gamithromycin was linked to specific farms; however, the practices associated with these differences in MIC classification were not determined from this analysis. Conditional probabilities for these associations can be found in Appendix A. No associations were detected between AMD treatment density and MIC classification of *P. multocida* isolates. 

#### 2.2.2. *P. multocida*: Animal-Level Variables

No animal-level variables included in this analysis were associated with the MIC classification of *P. multocida.* MIC classification was associated with the season during which the sample was collected and the source farm. The probability of non-susceptibility to florfenicol was higher in the hotter season (*p* = 67%) than in the cooler season (*p* = 52%). This relationship is mediated by other, unmeasured, farm-level factors, as demonstrated by the link between farm and florfenicol non-susceptibility in Figure 3. There is a complex pattern of non-susceptibility for the AMDs included in the study, including statistical dependencies between non-susceptibility to AMDs within the same class, including the macrolides gamithromycin, tildipirosin, and tilmicosin, as well as connections between the fluoroquinolones, danofloxacin and enrofloxacin. There are also statistical dependencies between AMDs across different classes, including enrofloxacin, danofloxacin, spectinomycin, and tulathromycin. Conditional probabilities for these associations can be found in Appendix A.

#### 2.2.3. *M. haemolytica*: Farm-Level Variables

Commercial and/or autogenous *Moraxella bovis +/− bovoculi* vaccination was associated with the MIC classification of *M. haemolytica* to enrofloxacin. The probability that an isolate was not susceptible to enrofloxacin, given that the farm vaccinated against pinkeye, was 63%, compared to 96% if the farm did not vaccinate against pinkeye. The non-susceptibility of *M. haemolytica* to danofloxacin and gamithromycin was linked to specific farms, as demonstrated in Figure 4. However, this analysis did not determine the practices associated with these differences in MIC classification. 

#### 2.2.4. *M. haemolytica*: Animal-Level Variables

No animal-level practices were linked to MIC classification of *M. haemolytica*. Non-susceptibility to gamithromycin, spectinomycin, and penicillin was associated with specific farms, as demonstrated in Figure 5. However, this analysis did not determine the animal-level practices associated with these differences. Conditional dependencies for both DAGs can be found in Appendix A.

#### 2.2.5. *H. somni*: Farm-Level Variables

Farm-level practices linked to MIC classification of *H. somni* include the number of source farms of a calf-rearing facility and respiratory vaccination practices. Vaccination with IRV1 was associated with the increased probability of an isolate being susceptible to tildipirosin, and this association was further influenced by the MIC classification of tilmicosin. The probability of tildipirosin non-susceptibility, given that a farm vaccinates with IRV1, was <1%, compared to 36% if the farm did not vaccinate with IRV1. However, if the isolate was also not susceptible to tildipirosin, the probability of tilmicosin non-susceptibility, given IRV1 vaccination, increased to 47%, compared to 5% if the farm vaccinated with IRV1, but the isolate was susceptible to tildipirosin. Linkages are demonstrated in Figure 6. 

#### 2.2.6. *H. somni*: Animal-Level Variables

A calf’s BRD score status was associated with tetracycline non-susceptibility, and this relationship was further influenced by phenicol class treatment history. The probability that an isolate was not susceptible to tetracycline, given that the calf was BRD score-positive, was 72%, compared to 54% if the calf was BRD score-negative. If the calf had received a phenicol treatment within 60 days of sampling, the probability of an isolate being not susceptible to tetracycline, given the calf was BRD-score-positive was 62%. The probability of an isolate being not susceptible to tetracycline, given the calf was BRD-score-negative, remained unchanged. Linkages are demonstrated in Figure 7. Conditional probabilities for these associations can be found in Appendix A.

## 3. Discussion

The current study findings have been used to generate hypotheses regarding animal- and farm-level practices that may be associated with MIC classification in the study population. It is important to note that, due to the cross-sectional study design and nature of these types of analyses, linkages do not infer causality. Instead, we suggest that linked variables are related in some way and may be useful for hypothesis generation or to aid in the direction of future investigations. The farm of sample origin appeared to be the most consistent factor related to MIC classification in both the BNA and logistic regression. While this association is due to underlying and unmeasured factors about the farm of sample origin, the management practices reported for farms that were associated with MIC classification can shed light on practices that may be related to AMR based on the analyses in this study.

In the logistic regression, farm of sample origin, feed lane cleaning method in heifer pens, vaccine practices, comingling of bull calves with heifer calves, injectable tetracycline treatment, and colostrum pasteurization were associated with MIC classification. The BNA explored the role of multiple related factors simultaneously and likewise demonstrated that the farm of sample origin was associated with MIC classification, suggesting that unmeasured practices are responsible for the differences, although non-susceptibility was not quantified and compared across farms using this analysis. The BNA likewise demonstrated associations between feed lane cleaning method, onsite milking, vaccination practices, and limited associations with AMD treatment history that may be related to MIC classification; the BNA additionally identified an association between the number of source farms and MIC classification.

Previous studies identified farm management practices as risk factors for disease, subsequent treatment with AMD, and AMR. Specifically, the use of lagoon water to flush alley ways in weaned pens on dairies has been shown to be associated with seropositivity against *Mycobacterium avium* subsp. *paratuberculosis* [16]. Furthermore, the use of lagoon water to flush below pre-weaned calves’ hutches has been associated with a higher prevalence of BRD [17]. The manure in lagoon water serves as a reservoir for infectious diseases, noxious gases that may influence health and immunity (e.g., ammonia that affects respiratory cilia), and is a possible route to spread AMR. 

The number of source farms may influence MIC classification by mixing calves from many different source farms and potentially increasing the spread of infectious disease early in life, increasing treatment for infectious diseases and thus increasing selective pressure for AMR in respiratory isolates. However, treatment history alone was only sometimes associated with MIC classification in this analysis; other factors associated with multiple source farms may be responsible, such as the possible spread of AMR genes from multiple source farms through the mixing of calves [18]. 

Vaccine practices may be associated with MIC classification due to differences in disease prevention and, thus, increased use of AMD when the disease is more common [19]. For example, *M. haemolytica* isolates from farms that vaccinated for pinkeye were more likely to be classified as susceptible to enrofloxacin; this may reflect more attention to disease prevention in weaned heifers. The link is likely indirect since enrofloxacin is not generally used as a treatment for pinkeye. However, this drug may be used to treat BRD, which may be misdiagnosed from signs of pinkeye (ocular discharge) or occur concurrently with pinkeye [17]. 

Hypotheses that may explain the association between the absence of onsite milking and increased classification of *P. multocida* as not susceptible to tildipirosin are less obvious. Of the four farms with onsite milking, two were single-source, and two were multisource, so it is unlikely that the farm type was truly responsible for differences in MIC classification. The fact that this association is mediated by the heifer feed lane cleaning method suggests a possible transfer of AMR from adult lactating cattle to the weaned heifers; however, it is unlikely that there is ecologic pressure to maintain tildipirosin AMR in adult dairy cattle because this drug is not used in adult dairy cattle due to the risk of milk residues. 

The BNA also revealed conditional dependencies between different AMD non-susceptibility outcomes. The association of AMR with drugs of the same class, such as those identified within the macrolide class and fluoroquinolone class, is not surprising since these drugs have similar mechanisms of action, and bacterial non-susceptibility mechanisms may be effective against more than one specific drug in a class. However, the linkages between MIC classification of drugs from different, unrelated classes, such as those identified between the fluoroquinolones and tulathromycin and spectinomycin, and tetracycline non-susceptibility with phenicol class AMD treatment history, suggest that bacterial mechanisms of non-susceptibility may be linked. Genetic links have been previously demonstrated in mobile genetic elements, and these elements may be circulating in the study population, as has been demonstrated with horizontal gene transfer in other cattle populations [20,21,22]. Previous studies have demonstrated that florfenicol non-susceptibility was genetically linked to tetracycline non-susceptibility, and persistence is likely related to co-selection [23,24,25,26].

Interestingly, neither individual animal treatment history nor an analysis of treatment frequency for a one-year cohort of calves on the same farms was consistently associated with MIC classifications in the respiratory isolates studied. These findings support the hypothesis that AMD use alone is not the only driver of AMR in the study population. Studies comparing AMR in enteric bacteria from beef or dairy cattle between herds that use or do not use AMD found minimal or variable differences in AMR, depending on what AMDs were investigated [27,28]. The observation that MIC classification is associated with the farm of sample origin suggests that non-susceptibility is a herd-level problem. This finding is not unique [23,29], but adds to the body of literature suggesting that farm-level management practices, and the interactions between host, pathogen, and environment [30,31,32,33], particularly in settings where animals are comingling, should be investigated for ways to decrease AMR in addition to the simple practice of limiting AMD use. From a clinical standpoint, simply using different AMDs to those with a high prevalence of non-susceptibility based on MIC testing may not change AMR outcomes as desired, particularly if AMR to unrelated classes of AMDs is genetically linked; co-selection may allow for selective pressure for AMR to classes of drugs not being used. The authors hypothesize that the evolutionary ecology in the study population, consisting not only of AMD use but also farm practices that spread or put other ecologic pressures on upper respiratory tract bacteria, maintains selective pressure for the MIC classifications observed in this study.

Season of sample collection was associated with MIC outcomes for some bacteria; *P. multocida* samples obtained in the hot season were associated with non-susceptibility to florfenicol in both the BNA and logistic regression. Conversely, *M. haemolytica* isolates obtained in the cool season were associated with non-susceptibility to macrolide AMDs. Seasonality has been associated with AMR outcomes in cattle enteric, human respiratory, and environmental studies [27,33,34]. The reasons why AMR may increase for some AMDs and decrease for others in the same season are unknown. However, it is possible as bacterial diseases of cattle can vary with seasons, and thus the bacterial ecology in the animal and in the environment may change seasonally; this may alter the environment in which selection pressures favor more or less AMR for AMDs of different classes.

The limitations of this study included relatively small naturally occurring treatment groups due to the retrospective nature of the records review. Additionally, only six farms were enrolled in the study; enrolling more farms could increase the ability to determine differences associated with farm management practices. An investigation of all possible factors that may have been involved in the maintenance or spread of AMR, such as host and pathogen genetics, unmeasured environmental factors, and additional unmeasured or unreported animal management factors, was beyond the scope of the study. Additionally, there was a lack of variability in some variables that limited the ability to make associations with these variables; for example, almost all *P. multocida* and *M. haemolytica* isolates were classified as non-susceptible tetracycline. Therefore, finding associations with tetracycline non-susceptibility in those isolates was unlikely. We conducted bootstrap analyses for the Bayesian networks that returned a graph with connections that we believe are non-spurious. However, with no known or validated model to compare our DAGs, some of the arcs may represent spurious associations. Similar analyses in other farms are needed to compare our DAG structure and strengthen our findings. Finally, some farm-level practices were practiced at only one farm in the study, making the effects of these variables impossible to untangle from other farm management practices.

This study reports associations between farm- and animal-level variables with AMR in respiratory bacterial isolates from weaned dairy heifers. Management factors at the farm level, such as the farm of sample origin and feed lane cleaning method were consistently associated with AMR across two methods of analysis. However, AMD treatment history was only variably associated with AMR outcomes. These findings add to the body of knowledge that suggests AMR is a herd-level problem and that herd management factors, other than AMD use alone, likely play important roles in the ecology of AMR on calf-rearing facilities. Most current efforts to decrease AMR center on decreasing AMD use; the findings of this study generate hypotheses involving animal management factors that may serve as additional control points for AMR on calf-rearing facilities. Further investigations into management factors related to the maintenance and spread of AMR in dairy calf-rearing facilities will be necessary to control AMR on calf-rearing facilities. 

## 4. Materials and Methods

A convenience sample of six California calf-rearing facilities were enrolled in a cross-sectional study that took place over two consecutive seasons: June 2019 (hot season) and February 2020 (cool season). The study was approved by the UC Davis Institutional Animal Care and Use Committee (protocol # 20114). Informed consent from herd management was obtained verbally prior to commencing study activities, as required by the Institutional Animal Care and Use Committee (IACUC) protocol for the study from which these data were obtained [4].

A sample size calculation was performed using a 2-sided test, a type 1 error of 0.05, a power of 80%, and an assumed percentage difference of 50% between the classification of AMR for BRD-score-positive compared to score-negative heifers. The BRD score was assigned as a binary outcome of either positive or negative for clinical signs of BRD, based on a clinical scoring system validated in weaned heifers [17]. A total of 283 heifers were required; however, to account for a 20% dropout rate, at least 340 heifers were required. To achieve similar sampling numbers between two seasons in all 6 farms, 360 animals were enrolled for sampling. Selection criteria included weaned heifers in group pens, ≥ 3 months of age, that had been comingled for at least 2 weeks before sampling, and that were less than 6 months of age based on farm records. Bull or steer calves comingled with heifers were excluded because they made up an inconsistent and small percentage of the population. Heifers scoring positive for BRD were enrolled in a convenience sample of all BRD-score-positive heifers in the pens available for sampling until 15 BRD-score-positive animals were identified per sampling time point. Heifers classified as BRD-score-negative were selected randomly, using a random number generator smartphone application, from the same pens until 15 BRD-score-negative animals were identified.

Deep nasopharyngeal swabs (DNPS) (double-guarded culture swabs, Reproduction Provisions LLC, Walworth WI, USA) were collected from enrolled heifers, as previously described [35]. Swabs were submitted for selective bacterial culture and AMD susceptibility determination for the following respiratory isolates: *Pasteurella multocida, Mannheimia haemolytica,* and *Histophilus somni.* Susceptibility testing included minimum inhibitory concentrations (MIC) determination using broth microdilution with Clinical Laboratory Standards Institute (CLSI) breakpoint analysis against 11 AMD (tetracycline, tilmicosin, tildipirosin, gamithromycin, enrofloxacin, danofloxacin, florfenicol, spectinomycin, tulathromycin, penicillin, ceftiofur) representing seven different drug classes (tetracycline, macrolide, fluoroquinolone, phenicol, aminocyclitol, penicillin, cephalosporin). Detailed methods of culture and sensitivity testing are provided in the Appendix A. The culture and sensitivity analysis results were previously published in a study reporting AMR prevalence in the study population [4]. The MIC values were classified as susceptible or not susceptible (intermediate or resistant) based on applicable Clinical & Laboratory Standards Institute (CLSI) breakpoints for *P. multocida, M. haemolytica,* or *H. somni* isolates to each specific AMD.

Farm-level variables were selected based on their potential influence on bacterial ecology, such as factors related to animal production classes, housing, feeding, and hygiene. Farm-level variables were obtained from farm management staff by conducting an in-person survey at the time of the first sampling for each farm and included: type of calf-rearing facility (single source or multisource, categorized by the number of sources); whether the calf-rearing facility has onsite milking; vaccines or biologicals administered to calves on-site; colostrum source; hospital milk feeding and pasteurization; whether bull calves are comingled with heifers; milk/grain/water group medication; feed lane cleaning method in weaned pens; and the season in which the sample was documented to account for the clustering of isolates during sampling periods. A sub-analysis was performed to evaluate the association between farm-level drug-use practices in weaned heifers preceding sampling and the MIC classification for *P. multocida* isolated from sampled calves. All AMD treatments from birth to six months of age were obtained for all calves received at or born on the farm over the year preceding the study sampling (1 January 2018–31 December 2018) for four of the six farms with one year of herd treatment data available for analysis, was obtained. Treatment frequency for this one-year cohort of calves was described as the number of AMD treatments by drug class, per 100 calves, for the drug classes penicillin, tetracycline, cephalosporin, macrolide, fluoroquinolone, sulfonamide, and phenicol. Treatment frequency was categorized as high if it was above the median value from the distribution of treatment density across the four farms or low if it was below this median value for each drug class.

Animal-level variables were selected based on their potential influence on bacterial ecology and these included the BRD score (positive or negative) and AMD treatment history. Calf breeds and BRD scores were recorded at the time of sampling. Treatment history (AMD) was obtained from electronic medical records, where available (n = 5/6 farms). One farm with missing medical records was excluded from the animal-level analyses. Treatment history was operationalized as days since treatment up to the sampling date for each AMD received and categorized, by drug class, into “never treated”, “treated less than 60 days prior to sampling”, and “treated more than 60 days prior to sampling”. For the BNA, any variables having less than 10 observations for a category were recategorized for all analyses. If recategorization was not possible, the variable was excluded from the analysis. 

### 4.1. Data Analysis 

#### 4.1.1. Logistic Regression

Mixed effects logistic regression analyses were performed to evaluate the association between farm- and animal-level variables and MIC classification to *P. multocida, M. haemolytica,* and *H. somni*, for each AMD tested. AMDs were grouped by class for the logistic regression analysis to limit the number of outcome-specific models needed for inference. Separate logistic regression models were developed for *P. multocida, M. haemolytica,* or *H. somni* and for farm-level and animal-level explanatory variables. Farm was considered a random effect, whereas other farm-level or animal-level variables were considered fixed effects. Stepwise forward selection of variables using Akaike Information Criterion (AIC) value to determine the best model fit was performed. The final model chosen was assessed for fitness using receiver operating characteristics (ROC) curve and area under the curve (AUC) values with a 95% confidence interval of the AUC. *p* < 0.05 was considered significant, and commercial statistical software was used for analysis (SAS v9.4 and JMP Pro v16.2, SAS Institute, Cary, NC, USA).

#### 4.1.2. Bayesian Network Analysis

Multilevel Bayesian networks with allow and block lists were used to represent the multilevel structure of the data [36]. Block lists define which node connections should be denied or blocked in the network and allow lists are forced connections between nodes in the network. Forcing a connection between season and “Farm” accounted for the clustering of farm samples within each season. Blocking all connections from farm- and animal-level variables to “Farm” accounted for the nested structure of the data [36]. 

Separate models were built for each respiratory pathogen using a purely data-driven, exploratory approach. Antimicrobial drugs were not grouped by class for the BNA and all variables included in the models were categorical. The steps for BNA have been described previously [37]. Briefly, the analysis involves (i) structure learning using the score-based hill-climbing (HC) learning algorithm, which determines the topological relationships between the nodes in the network and the sample data [38]; (ii) bootstrapping using 10,000 bootstrap samples to prune connections observed in less than 50% of the bootstrap samples [10,37,39,40]; and (iii) parameter learning using the Bayesian method to estimate the conditional probabilities between connected nodes from the identified network and the observed sample data [41].

To visualize the results from the BNA, DAGs were generated. To ensure clarity for readers, statistical dependencies between non-susceptibility, as determined by MIC classification to several AMDs, and the farm- and animal-level variables, any node with only a singular, forced connection to “Farm” (i.e., no statistical relationship to MIC classification of not susceptible for any AMDs) was removed from the manuscript DAG to streamline the presentation. Further, due to the cross-sectional nature of the data and the inability to determine temporality in AMD susceptibility, causality was not inferred from the DAG. Therefore, the arc direction for the DAGs in the manuscript was removed for simplicity of interpretation. The original, unaltered DAGs can be viewed in the Appendix A. All analyses were conducted in R using the package *bnlearn* [42].

## Figures and Tables

**Figure 1 antibiotics-13-00050-f001:**
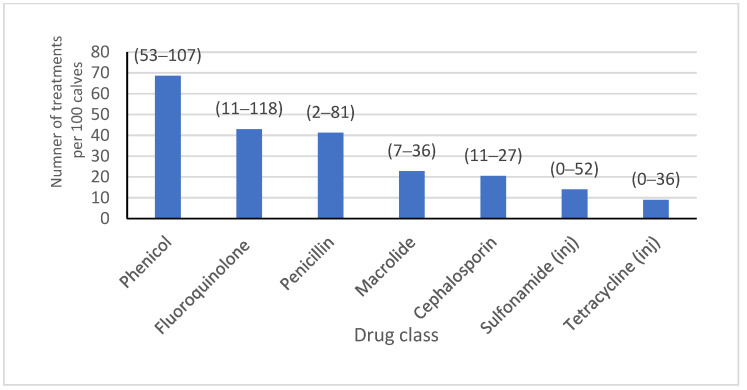
Farm-level treatment density. The treatment frequency of AMDs administered individually and parenterally to all calves ages 0–180 days over the course of 1 year of calf births or arrivals for 4 calf-rearing facilities in California. Treatment frequency is displayed as the average number of individual animal parenteral AMD drug treatments, per 100 calves, by AMD class. Range of treatments per 100 calves across farms is reported in parentheses above each column. Medication in feed or water of animal groups is not included in this figure.

**Figure 2 antibiotics-13-00050-f002:**
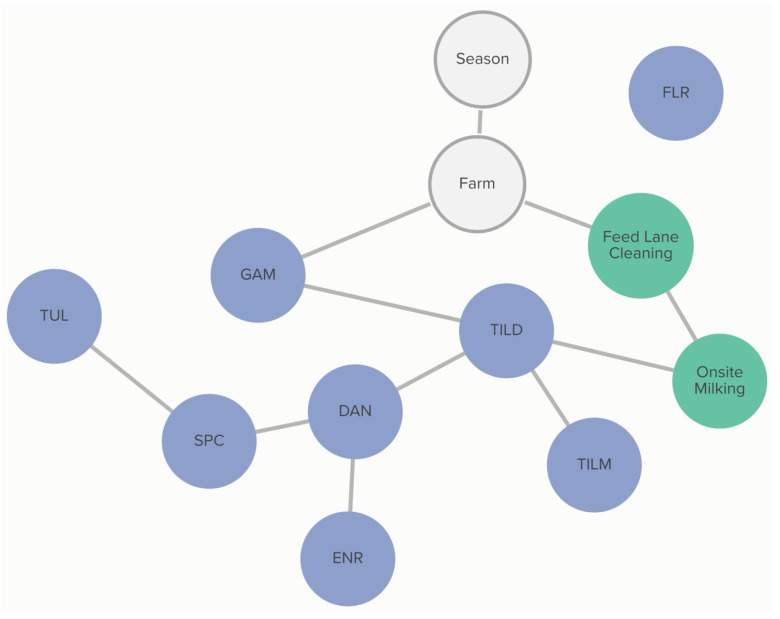
Graph of farm-level variable relationships with *P. multocida* MIC classification from Bayesian network analysis. Linkages between *P. multocida* MIC classification to AMDs and farm-level risk factors across 6 farms for isolates and risk factors obtained from weaned heifers in California. Green nodes are farm-level variables; blue nodes are AMD non-susceptibility classification variables; grey nodes are clustering variables. Only farm-level variables connected to AMD MIC classification are presented here. Tilmicosin (TILM), tildipirosin (TILD), gamithromycin (GAM), danofloxacin (DAN), enrofloxacin (ENR), florfenicol (FLR), tulathromycin (TUL), and spectinomycin (SPC).

**Figure 3 antibiotics-13-00050-f003:**
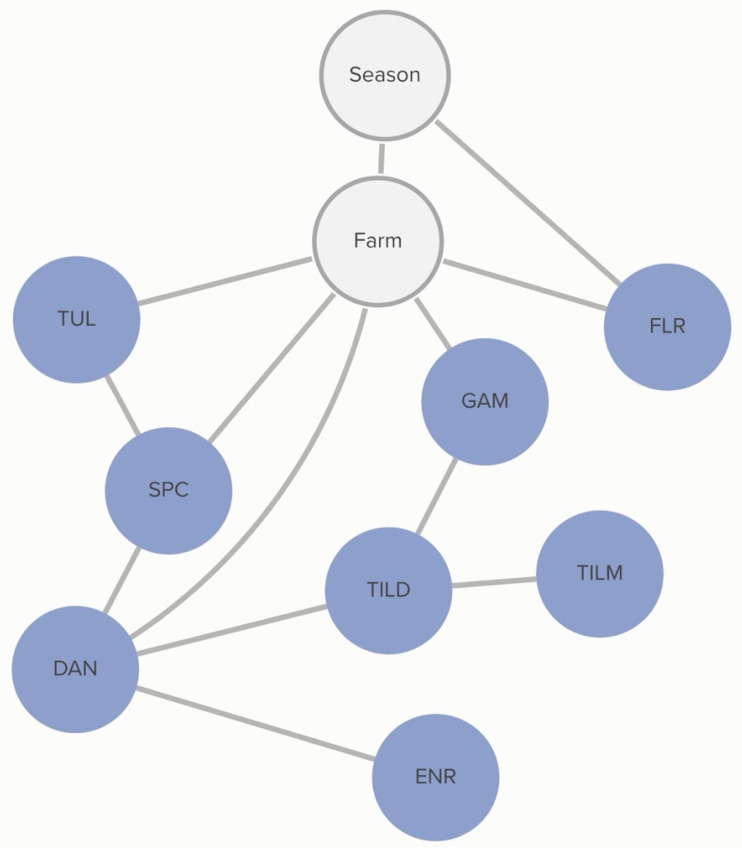
Graph of animal-level variable relationships with *P. multocida* MIC classification from the Bayesian network analysis. Linkages between *P. multocida* MIC classification to AMDs and animal-level risk factors across 5 farms for which animal treatment data were available, for isolates and risk factors obtained from weaned heifers in California. Green nodes are animal-level variables; blue nodes are AMD non-susceptibility classification variables; grey nodes are clustering variables. Only animal-level variables connected to AMD MIC classification are presented here. Tulathromycin (TUL), tilmicosin (TILM), tildipirosin (TILD), gamithromycin (GAM), florfenicol (FLR), danofloxacin (DAN), enrofloxacin (ENR), and spectinomycin (SPC).

**Figure 4 antibiotics-13-00050-f004:**
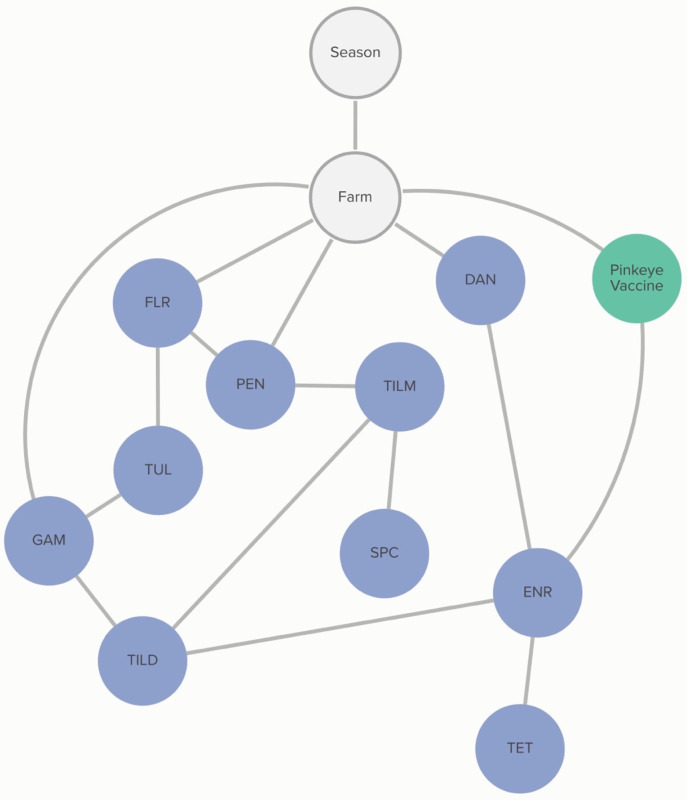
Graph of farm-level variable relationships with *M. haemolytica* MIC classification from Bayesian network analysis. Linkages between *M. haemolytica* MIC classification to AMDs and farm-level risk factors across 6 farms for isolates and risk factors obtained from weaned heifers in California. Green nodes are farm-level variables; blue nodes are AMD non-susceptibility classification variables; grey nodes are clustering variables. Only farm-level variables connected to AMD MIC classification are presented here. Tulathromycin (TUL), tilmicosin (TILM), tildipirosin (TILD), tetracycline (TET), gamithromycin (GAM), penicillin (PEN), florfenicol (FLR), danofloxacin (DAN), enrofloxacin (ENR), and spectinomycin (SPC).

**Figure 5 antibiotics-13-00050-f005:**
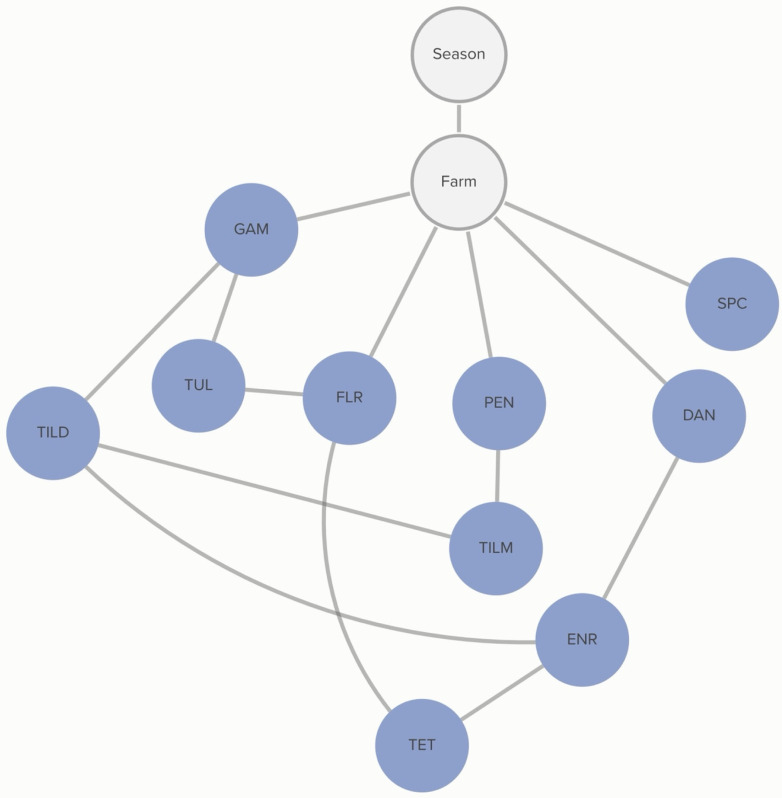
Graph of animal-level variable relationships with *M. haemolytica* MIC classification from Bayesian network analysis. Linkages between *M. haemolytica* MIC classification to AMDs and animal-level risk factors across 5 farms for which animal treatment data were available for isolates and risk factors obtained from weaned heifers in California. Green nodes are animal-level variables; blue nodes are AMD non-susceptibility classification variables; grey nodes are clustering variables. Only animal-level variables connected to AMD MIC classification are presented here. Penicillin (PEN), tulathromycin (TUL), tilmicosin (TILM), tildipirosin (TILD), tetracycline (TET), gamithromycin (GAM), florfenicol (FLR), danofloxacin (DAN), enrofloxacin (ENR), and spectinomycin (SPC).

**Figure 6 antibiotics-13-00050-f006:**
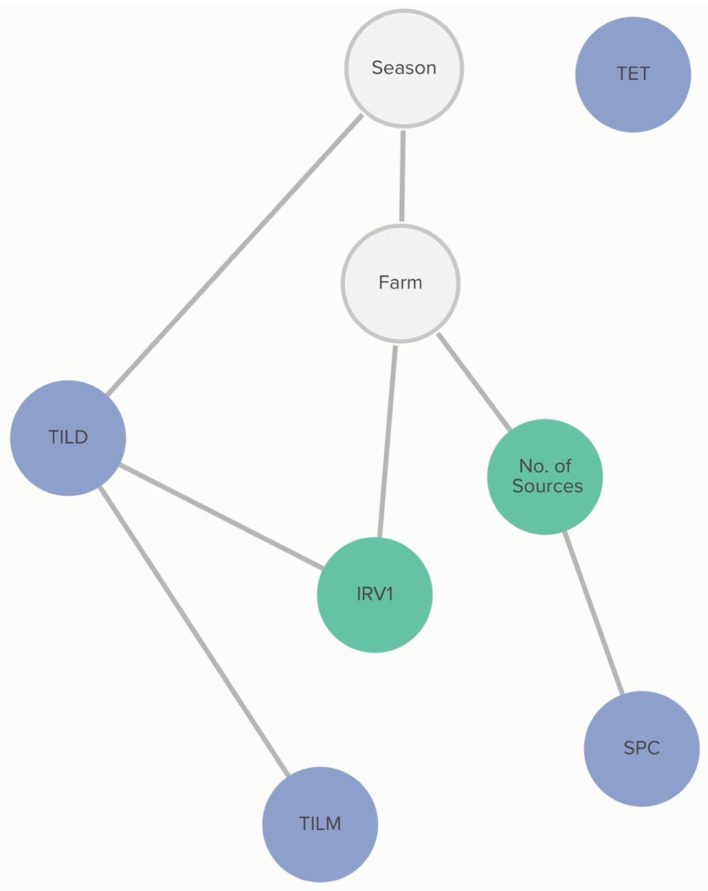
Graph of farm-level variable relationships with *H. somni* MIC classification from Bayesian network analysis. Linkages between *H. somni* MIC classification to AMDs and farm-level risk factors across 6 farms for isolates and risk factors obtained from weaned heifers in California. Green nodes are farm-level variables; blue nodes are AMD non-susceptibility classification variables; grey nodes are clustering variables. Only farm-level variables connected to AMD MIC classification are presented here. Modified live injectable respiratory vaccine 1 (IRV1), tilmicosin (TILM), tildipirosin (TILD), tetracycline (TET), and spectinomycin (SPC).

**Figure 7 antibiotics-13-00050-f007:**
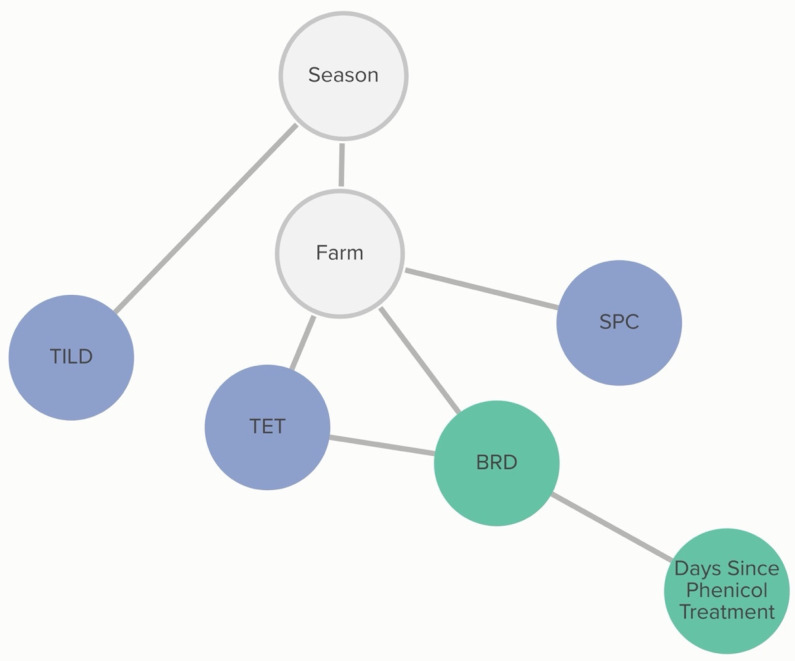
Graph of animal-level variable relationships with *H. somni* MIC classification from Bayesian network analysis. Linkages between *H. somni* MIC classification to AMDs and animal-level risk factors across 5 farms for which animal treatment data were available for isolates and risk factors obtained from weaned heifers in California. Green nodes are animal-level variables; blue nodes are AMD non-susceptibility classification variables; grey nodes are clustering variables. Only animal-level variables connected to AMD MIC classification are presented here. Tildipirosin (TILD), tetracycline (TET), and spectinomycin (SPC).

**Table 1 antibiotics-13-00050-t001:** Summary statistics. A summary of farm-level variables for all isolates included in the farm-level analysis (n = 361 isolates from n = 341 calves). The number of farms practicing each variable is included in parenthesis following the variable name. AMD = antimicrobial drug; tetracycline (TET), tilmicosin (TILM), tildipirosin (TILD), gamithromycin (GAM), enrofloxacin (ENR), danofloxacin (DAN), florfenicol (FLR), spectinomycin (SPC), tulathromycin (TUL), penicillin (PEN), ceftiofur (CEF).

Variable	n (%)
Farm	
1	47 (13.0)
2	61 (16.9)
3	58 (16.1)
4	69 (19.1)
5	65 (18.1)
6	61 (16.9)
Bacteria Cultured	
*P. multocida*	145 (40.2)
*M. haemolytica*	119 (33.0)
*H. somni*	97 (26.9)
Not Susceptible Classification for AMDs	
TET	164 (45.4)
TILM	199 (55.1)
TILD	188 (52.1)
GAM	148 (41.0)
ENR	175 (48.5)
DAN	173 (47.9)
FLR	126 (34.9)
SPC	122 (33.8)
TUL	60 (16.6)
PEN	45(12.5)
CEF	0 (0.0)
Season Sampled	
Hot season	174 (48.2)
Cool season	187 (51.8)
Type of Calf-Rearing Facility	
Single source (2/6)	130 (36.0)
Multiple sources, less than 10 (2/6)	108 (29.9)
Multiple sources, more than 10 (2/6)	123 (34.1)
Farm has Onsite Milking	
Yes (4/6)	256 (70.9)
No (2/6)	105 (29.1)
Vaccines Administered	
Intranasal respiratory vaccine (5/6)	296 (82.0)
*Salmonella* vaccine (3/6)	167 (46.2)
Modified live injectable respiratory vaccine 1 (3/6)	173 (47.9)
Modified live injectable respiratory vaccine 2 (2/6)	119 (33.0)
Modified live injectable respiratory vaccine 3 (1/6)	69 (19.1)
Pinkeye vaccines (4/6)	249 (69.0)
Commercial clostridial combination vaccine (1/6)	69 (19.1)
Colostrum Source	
Pooled (5/6)	295 (81.7)
From Dam (1/6)	66 (18.3)
Hospital Milk	
Pasteurized (5/6)	296 (82.0)
Not pasteurized (1/6)	65 (18.0)
Total Solids Screening	
Screen only (5/6)	303 (83.9)
Premium paid based on total solids (1/6)	58 (16.1)
Bull Calves	
Co-mingled with heifers (3/6)	177 (49.0)
Not comingled with heifers (3/6)	184 (50.1)
Antimicrobial Drug Treatment	
Farm medicates milk in groups (3/6)	173 (47.9)
Farm medicates grain with penicillin in groups (1/6)	47 (13.0)
Farm medicates grain with amprolium in groups (3/6)	184 (51.0)
Farm medicates grain with monensin in groups (5/6)	314 (87.2)
Farm medicates water with a tetracycline in groups (5/6)	300 (83.1)
Farm medicates water with a sulfonamide in groups (4/6)	235 (65.1)
Feed Lane Cleaning Method	
Scrape (2/6)	105 (29.1)
Lagoon Flush (2/6)	122 (33.8)
Clean Water Flush (1/6)	69 (19.1)
Scrape & Lagoon Flush (1/6)	65 (18.0)
Dust Management	
Yes (5/6)	296 (82.0)
No (1/6)	65 (18.0)

**Table 2 antibiotics-13-00050-t002:** Summary statistics. A summary of animal-level variables for all isolates included in the animal-level analysis (n = 296 isolates from n = 276 calves). *P. multocida* (n = 121); *M. haemolytica* (n = 110); *H. somni* (n = 65); AMD = antimicrobial drug. Tetracycline (TET), tilmicosin (TILM), tildipirosin (TILD), gamithromycin (GAM), enrofloxacin (ENR), danofloxacin (DAN), florfenicol (FLR), spectinomycin (SPC), tulathromycin (TUL), penicillin (PEN), ceftiofur (CEF). Samples from farm 5 were excluded due to a lack of records available for analysis. Animal-level variables were categorized so that each category had at least 10 observations. If it was not possible to recategorize a variable to meet this requirement, the variable was excluded from the analysis (denoted by grey in the table).

Variable	n (%)		
	*P. multocida*	*M. haemolytica*	*H. somni*
Farm			
1	13 (10.7)	23 (20.9)	11 (16.9)
2	24 (19.0)	26 (23.6)	11 (16.9)
3	21 (17.4)	17 (15.5)	20 (30.8)
4	34 (28.9)	20 (18.2)	15 (23.1)
5		excluded	
6	29 (24.0)	24 (21.8)	8 (12.3)
Season sampled			
Hot season	61 (50.4)	55 (50.0)	27 (41.5)
Cool season	60 (49.6)	55 (50.0)	38 (58.5)
Not Susceptible Classification for AMDs			
TET		95 (86.4)	42 (64.6)
TILM	92 (76.0)	68 (61.8)	
TILD	89 (73.6)	55 (50.0)	15 (23.1)
GAM	84 (69.4)	39 (35.5)	
ENR	70 (57.9)	76 (69.1)	
DAN	70 (57.9)	75 (68.2)	
FLR	72 (59.5)	38 (34.5)	
SPC	33 (27.3)	17 (15.5)	43 (66.2)
TUL	23 (19.0)	32 (29.1)	
PEN		40 (36.4)	
CEF			
Total number of AMD treatments			
0	21 (17.4)	30 (27.3)	35 (53.8)
1 to 2	34 (28.1)	35 (31.8)	
3 or more	66 (54.5)	45 (40.9)	30 (46.2)
Last AMD treatment			
Never treated	21 (17.4)	29 (26.4)	
<16 days	19 (15.7)	14 (12.7)	
16-60 days	33 (27.3)	25 (22.7)	
≥60 days	48 (39.6)	42 (38.2)	
Days since penicillin treatment			
Never treated	94 (77.7)	92 (83.6)	47 (72.3)
<60 days	17 (14.0)	18 (16.4)	18 (27.7)
≥60 days	10 (8.3)		
Days since cephalosporin treatment			
Never treated	101 (83.5)	100 (90.9)	
<60 days	10 (8.3)	10 (9.1)	
≥60 days	10 (8.3)		
Days since macrolide treatment			
Never treated	82 (67.8)	86 (78.2)	47 (72.3)
<60 days	12 (9.9)	24 (21.8)	18 (27.7)
≥60 days	27 (22.3)		
Days since phenicol treatment			
Never treated	52 (43.0)	52 (47.3)	27 (41.5)
<60 days	27 (22.3)	24 (21.8)	12 (18.5)
≥60 days	42 (34.7)	34 (30.9)	26 (40.0)
Days since tetracycline treatment			
Never treated	96 (79.3)	95 (86.3)	52 (80.0)
<60 days	25 (20.7)	15 (13.6)	13 (20.0)
≥60 days			
Days since sulfonamide treatment			
Never treated	101 (83.5)		
<60 days	20 (16.5)		
≥60 days			
Days since fluoroquinolone treatment			
Never treated	78 (64.5)	81 (73.6)	48 (73.8)
<60 days	14 (11.6)	13 (11.8)	17 (26.2)
≥60 days	29 (24.0)	16 (14.5)	

**Table 3 antibiotics-13-00050-t003:** Logistic regression results. A summary of logistic regression models predicting MIC classification of not susceptible for *P. multocida* (n = 121); *M. haemolytica* (n = 110); *H. somni* (n = 65) isolates as a function of significant (*p* < 0.05) farm- and animal-level variables; AMD = antimicrobial drug; fluoroquinolone (FLQ), florfenicol (FLR), macrolide (MAC), spectinomycin (SPC), sulfonamides (SUL), oxytetracyline (OXY), penicillin (PEN), tetracycline (TET).

	AMD	Parameter Estimate	*p*-Value
*P. multocida*			
Farm-level variables			
No IRV1 vaccination	FLQ	0.95	<0.0001
No IRV3 vaccination	FLQ	−0.93	<0.0001
No IRV3 vaccination	FLR	−2.49	<0.0001
No IRV3 vaccination	MAC	−1.79	<0.0001
Scraping feed lane and clean water flushing compared to scraping and lagoon water flushing	SPC	1.58	<0.0001
Animal variables			
No previous treatment with TET	FLQ	−0.89	0.01
No previous treatment with SUL	FLQ	−0.82	0.01
No previous treatment with OXY	FLR	−1.69	<0.0001
Season sampled (hot compared to cool)	FLR	−0.49	0.03
Positive BRD score	MAC	0.80	0.02
No previous treatment with SUL	MAC	−1.50	<0.0001
*M. hemolytica*			
Farm variables			
Scraping feed lane and clean water flushing compared to scraping and lagoon water flushing	FLR	1.01	<0.0001
Scraping feed lane and clean water flushing compared to scraping and lagoon water flushing	PEN	1.27	<0.0001
No comingling bull calves and heifers	SPC	−1.37	0.0004
No IRV1 vaccination	TET	1.05	0.007
No comingling bull calves and heifers	MAC	−0.72	0.001
Group medication in water	MAC	0.56	0.02
Animal variables			
No previous treatment with TET	FLQ	−1.92	0.0003
Season sampled (hot compared to cool)	FLR	−0.60	0.01
No previous treatment with TET	FLR	0.85	0.01
No previous treatment with FLQ	FLR	0.67	0.007
No previous treatment with TET	PEN	1.75	0.001
Season sampled (hot compared to cool)	MAC	0.42	0.042
No previous treatment with MAC	MAC	0.91	0.006
*H. somni* ^†^			
Farm variables			
Colostrum pasteurization	TET	0.92	0.03
No IRV1 vaccination	TET	0.62	0.002

^†^ There was no association between animal-level variables and the MIC classification of not susceptible for *H*. *somni* to tetracyclines or macrolides.

## Data Availability

This study was funded by the Antimicrobial Use and Stewardship (AUS) Program of the California Department of Food and Agriculture (CDFA) and is subject to California Food and Agricultural Code (FAC) Sections 14400 to 14408. FAC Section 14407 requires that data collected be held confidential to prevent the individual identification of a farm or business; as such, only aggregated data from this study can be shared and raw data are not able to be shared.

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
