# Peer review of "Antimicrobial Susceptibility in Respiratory Pathogens and Farm and Animal Variables in Weaned California Dairy Heifers: Logistic Regression and Bayesian Network Analyses"

_antibiotics, 2024, doi:10.3390/antibiotics13010050_

Round 1

Reviewer 1 Report

Comments and Suggestions for Authors

Associations between antimicrobial susceptibility in respiratory pathogens and farm and animal variables in weaned Calfornia dairy heifers using logistic regression and Bayesian network analyses

Finding a correlation between AMR and other factors is an interesting field of research. The article will help the researchers to use different variables, such as farm or animal variables, in their future research regarding AMR associations with different variables. The article is well written; however, it sometimes contains complex / longer sentences that may be difficult to understand for the readers. The following are some suggestions to improve the manuscript for better understanding by the reader.

The abstract certainly prepares the reader for an interesting research idea. However, the inclusion of a specific aim and some key results may help to increase the interest of the readers. The abstract may include one line of conclusion for the study at the end.

Line25

Bovine Respiratory disease (BVD)

Line-28

Some of these variable may be mentioned here

Line-30

Type of data?

Line-34

AMDs

Line-26

Introduction: The beginning of the introduction can be improved by keeping sentence length shorter to create a stronger connection with readers.

Line-62

condition

Line-70

Furthermore,

Line-73

AMDs

Line-78

California

Line-84

"Not susceptible"

Can we use resistant?

Does it include resistant and intermediate

Line-99---

The sentence can be shortened or divided into two sentences. There are a few other examples, too, that can be managed.

Line-100, 101, 125, Table-1

Italicize "Salmonella"

Line-155

omit AMD

Line-Table-2

Not susceptible Classification or AMDs

Line-176-177

Line can be repharased!!

Line-179

IRV3 has already been described as

Commercial modified live injectable respiratory viral combination vaccine 3 (IRV3)

only IRV3 may be sufficient later on

Line-193

MIC classification?

Line-419-440, 449-472

Discussion lacks any reference. 

Results may be discussed with reference to previous studies. two these references are mentioned but author can look for other references too.

Anwar MA, Aziz S, Ashfaq K, Aqib AI, Shoaib M, Naseer MA, Alvi MA, Muzammil I, Bhutta ZA, Sattar H, Saleem A, Zaheer T, Khanum F and Mahmood A, 2022. Trends in frequency, potential risks, and antibiogram of E. coli isolated from semi-intensive dairy systems. Pak Vet J, 42(2): 167-172. http://dx.doi.org/10.29261/pakvetj/2022.018

Sebbar G, Fellahi S, Filali-Maltouf A and Belkadi B, 2021. Detection of colistin resistance in Mannheimia haemolytica & Pasteurella multocida isolates from ruminants in Morocco. Pak Vet J, 41(1): 127-131. http://dx.doi.org/10.29261/pakvetj/2020.077

Line-447

noxious gasses?

Line-517

The last paragraph of the discussion may be based on "conclusion" rather than "limitations."

Line-573

The first animal-level variable mentioned here is "BREED".

However, we will not find any information or analysis regarding breed.

Comments on the Quality of English Language

Some sentences can be shortened for better understanding.

Some corrections have been suggested.

Author Response

Line25

Bovine Respiratory disease (BVD)

We added bovine to this line at 24-25

Line-28

Some of these variable may be mentioned here

We added an example of some of the variables considered at line 28

Line-30

Type of data?

We added “survey” to line 30

Line-34

AMDs

Thank you for catching this, we updated line 34 to AMDs

Line-26

Introduction: The beginning of the introduction can be improved by keeping sentence length shorter to create a stronger connection with readers.

We broke this line (24-26) into two lines by adding a period where the semicolon was previously (line 24)

Line-62

Condition

We changed the text in line 62 from “disease system” under study to “condition” under study.

Line-70

Furthermore,

We added “Furthermore,” before the start of the new sentence on line 70

Line-73

AMDs

Thank you for catching this, we changed AMD to AMDs on line 73

Line-78

California

We added “California” in line 78 to be more explicit about the location of the weaned dairy heifers.

Line-84

"Not susceptible"

Can we use resistant? Does it include resistant and intermediate

We chose not to use “resistant” because in this study we grouped resistant and intermediate together (line 594). So, the accurate interpretation is susceptible and not susceptible.

Line-99---

The sentence can be shortened or divided into two sentences. There are a few other examples, too, that can be managed.

We agree with the reviewer that this is a very long sentence. To make it more digestible to readers but not sacrifice efficiency in listing the vaccines and biologicals used we removed “three types, containing combinations of” at line 101, removed the repetitive use of “vaccine(s) after each vaccine type, and removed “inconsistent” from line 104. Finally, we split the sentence in two at line 106.

Line-100, 101, 125, Table-1

Italicize "Salmonella"

We corrected “Salmonella” in these instances on lines 100, 101, 125, and in Table 1

Line-155

omit AMD

We removed “AMD” after cephalosporin on line 155

Line-Table-2

Not susceptible Classification or AMDs

We added “for AMDs” after “Not susceptible classification” in Table 2 and also in Table 1 for consistency.

Line-176-177

Line can be repharased!!

We removed “MIC classification of susceptible for” and simplified the sentence by adding “being susceptible” to fluoroquinolones at lines 179-180

Line-179

IRV3 has already been described as Commercial modified live injectable respiratory viral combination vaccine 3 (IRV3) only IRV3 may be sufficient later on

We removed “commercial modified live injectable respiratory viral combination vaccine” from this line (181) and just left IRV3

Line-193

MIC classification?

We changed was associated with “MIC classification of” to was associated with “susceptibility” of at line 195.

Line-419-440, 449-472

Discussion lacks any reference. 

Results may be discussed with reference to previous studies. two these references are mentioned but author can look for other references too.

We appreciate the reviewer’s detailed review of the manuscript and thoughtful comments/recommendations. The beginning of the discussion section is a summary of the results to frame the following discussion points. There are no references included in lines 429-450 because it is a recap of our findings. We understand lines 459-481 as an area where we discuss and contextualize our results without references. We have added references to this section on lines 465, 467, 473, and further down on line 503. Based on other comments, we did add a new section to the discussion discussing our seasonality findings with reference to previous studies that can be found on lines 517-526.

Line-447

noxious gasses?

We added an example of noxious gas after this term on lines 457-458.

Line-517

The last paragraph of the discussion may be based on "conclusion" rather than "limitations."

Thank you for bringing this up, we agree it’s important to end the paper with a conclusion rather than limitations and have added a conclusion paragraph at lines 545-556.

Line-573

The first animal-level variable mentioned here is "BREED". However, we will not find any information or analysis regarding breed.

Thank you for catching this – we removed breed because of little variability in breed type (an overwhelming majority were Holstein) and no differences or associations detected by breed. We removed “breed” from line 573 to avoid any confusion for readers.

Reviewer 2 Report

Comments and Suggestions for Authors

Please shorten the title like “Antimicrobial Susceptibility in Respiratory Pathogens and Farm/Animal Variables in Weaned California Dairy Heifers: Logistic Regression and Bayesian Network Analysis

Methods and results section:

1. How was verbal informed consent obtained from herd management, and was it documented in any way?

2. Can you provide the specific values used for the parameters in the sample size calculation, such as the type 1 error rate, power, and the assumed percentage difference?

3. What was the rationale behind the selection criteria for heifers, and why were bull or steer calves excluded from the study?

4. Could you provide more details about the process of collecting deep nasopharyngeal swabs (DNPS), including the number of swabs taken per heifer and storage conditions?

5. How were the MIC values classified as susceptible or not susceptible, and what is the significance of this classification for each specific antimicrobial drug (AMD)?

6. Can you explain the relevance of the farm-level variables, such as the type of calf rearing facility and the presence of onsite milking, in the context of your study?

7. For animal-level variables like breed, BRD score, and AMD treatment history, how do these factors relate to the outcomes you're investigating?

8. What is the purpose of Bayesian Network Analysis (BNA) in your study, and how will you visually represent the results of BNA?

9. How do you plan to handle and present the complex relationships identified by Bayesian Network Analysis (BNA) to ensure clarity for readers?

10. Why is it important to emphasize that causality is not inferred from the Directed Acyclic Graphs (DAGs) due to the cross-sectional nature of the data?

10. Results: did the study consider potential interactions between variables that could have been missed in the analysis?

11. Were there any outliers or extreme data points that could have had a disproportionate impact on the results?

12. Did the study consider potential interactions between variables that could have been missed in the analysis? 

Author Response

Please shorten the title like “Antimicrobial Susceptibility in Respiratory Pathogens and Farm/Animal Variables in Weaned California Dairy Heifers: Logistic Regression and Bayesian Network Analysis”

We removed “Associations between” and “using” from the title to shorten it. We did leave “farm and animal” because it is more grammatically formal than “farm/animal”

Methods and results section:

  1. How was verbal informed consent obtained from herd management, and was it documented in any way?

The Institutional Animal Care and Use Committee (IACUC) protocol for the study for which these data were obtained stipulated that informed consent be obtained. The principal investigator personally obtained informed verbal consent from farm management before commencing daily study activities. We added this information about IACUC at lines 562-564 and included a citation to the original study from which these data were obtained.

  1. Can you provide the specific values used for the parameters in the sample size calculation, such as the type 1 error rate, power, and the assumed percentage difference?

This information was provided on lines 565-567.

  1. What was the rationale behind the selection criteria for heifers, and why were bull or steer calves excluded from the study?

Bull calves were excluded from the study because they made up an inconsistent and small percentage of the population. We added this information in lines 575-576.

  1. Could you provide more details about the process of collecting deep nasopharyngeal swabs (DNPS), including the number of swabs taken per heifer and storage conditions?

We provide a citation to a paper that outlines the protocols followed for the DNPS in more detail than we can in this paper at line 583.

  1. How were the MIC values classified as susceptible or not susceptible, and what is the significance of this classification for each specific antimicrobial drug (AMD)?

We provide detailed methods of the culture and sensitivity testing in the supplemental materials (lines 715-744). We added that these were classified based on CLSI breakpoints in lines 594-595.

  1. Can you explain the relevance of the farm-level variables, such as the type of calf-rearing facility and the presence of onsite milking, in the context of your study?

We chose farm-level variables that may influence the bacterial ecology in heifer-rearing environments, we added this information to lines 558-559.

  1. For animal-level variables like breed, BRD score, and AMD treatment history, how do these factors relate to the outcomes you're investigating?

Similar to above, we chose animal-level variables that may influence the bacterial ecology in heifer-rearing environments and added this information to lines 618-619.

  1. What is the purpose of Bayesian Network Analysis (BNA) in your study, and how will you visually represent the results of BNA?

We discuss the reasons for using BNA in the study in the introduction on lines 60-66 and 70-73. We also mention how the results will be visualized from the BNA on lines 68-69. We added a reminder that results are visually represented using DAGs in the methods section on line 662.

  1. How do you plan to handle and present the complex relationships identified by Bayesian Network Analysis (BNA) to ensure clarity for readers?

To ensure clarity and ease of interpretation for readers, we simplified the presentation of the DAGs by removing nodes with only a singular, forced connection to the farm variable (having no statistical relationship with any of the AMDs under study). We mention this on lines 624-628. We added a note that this is to ensure clarity for readers at lines 662-663.

  1. Why is it important to emphasize that causality is not inferred from the Directed Acyclic Graphs (DAGs) due to the cross-sectional nature of the data?

We added that causality cannot be inferred from the DAGs because the cross-sectional nature of the data inhibits our ability to determine temporality in drug susceptibilities on lines 667-668.

  1. Results: did the study consider potential interactions between variables that could have been missed in the analysis?

(See response to 12)

  1. Were there any outliers or extreme data points that could have had a disproportionate impact on the results?

All variables in the study were categorical and thus there were no outliers or extreme data points. We mention that all variables are categorical on lines 653-654.

  1. Did the study consider potential interactions between variables that could have been missed in the analysis? 

The authors agree that certainly there are additional variables that may play a role in AMR on a calf rearing facility; the finding that farm of sample origin was consistently associated with AMR outcomes, while specific management factors were variably associated with outcomes suggests that there are unmeasured variables that interact with AMR. The introductory paragraph of the discussion poses this consideration to the reader as follows, “While this association is due to underlying and unmeasured factors about the farm of sample origin, the management practices reported for farms that were associated with MIC classification can shed light on practices that may be related to AMR based on the analyses in this study”. We have added a line has been added to the discussion on lines 530-533 to clarify this important point as follows, “Investigation of all possible factors that may be involved in the maintenance or spread of AMR, such as host and pathogen genetics, unmeasured environmental factors, and additional unmeasured or unreported animal management factors, was beyond the scope of the study” 

Reviewer 3 Report

Comments and Suggestions for Authors

1. Line 171: >=60 days data is missing for H.somni.

2. There are so many values written in text, I would recommend that the authors should present the data in some other form such as graphs or charts or by other means so that it is easy to understand.

3. Data presentation can be improved.

Comments on the Quality of English Language

The manuscript language and grammar is understandable. 

Author Response

  1. Line 171: >=60 days data is missing for H.somni.

This is due to an issue with the table splitting pages but the >=60 days for H.somni had to be regrouped and that box is actually part of <60 days so it’s 17(26.2) should span those two (so the variable essentially became never treated or treated). The table is no longer split between two pages and should be correct.

  1. There are so many values written in text, I would recommend that the authors should present the data in some other form such as graphs or charts or by other means so that it is easy to understand.

Please see response to point 3

  1. Data presentation can be improved.

The Bayesian network analysis results are presented both in text and graphically. We added a table to present the results from the logistic regression analysis so that readers can look over the findings in a more user-friendly way in addition to the in-text results. This new table (Table 3) was entered at line 297.

Reviewer 4 Report

Comments and Suggestions for Authors

The authors used logistic regression and BNA to describe associations between farm and animal level variables and AMR in respiratory bacterial isolates  They included in their studies 145, 119, and 97 P. multocida, M. haemolytica, and H. somni (respectively) isolates (n = 361) collected  from 341 weaned dairy heifers on six farms 31 in California.  They analyzed inhibitory concentration (MIC) classification of respiratory isolates against 11 antimicrobial drug. It is interesting studies with a lots of  date including vaccines administered, antimicrobial drug treated.  The concluded that:” Animal level variables associated with a MIC classification included whether the calf was respiratory disease (BRD) score positive and time since the last phenicol treatment”. The discussion is very extensive, so I think there should be a Conclusion section.     

Author Response

The authors used logistic regression and BNA to describe associations between farm and animal level variables and AMR in respiratory bacterial isolates. They included in their studies 145, 119, and 97 P. multocida, M. haemolytica, and H. somni (respectively) isolates (n = 361) collected from 341 weaned dairy heifers on six farms 31 in California.  They analyzed inhibitory concentration (MIC) classification of respiratory isolates against 11 antimicrobial drug. It is interesting studies with a lots of  date including vaccines administered, antimicrobial drug treated.  The concluded that:” Animal level variables associated with a MIC classification included whether the calf was respiratory disease (BRD) score positive and time since the last phenicol treatment”. The discussion is very extensive, so I think there should be a Conclusion section.     

We appreciate the reviewers' comments and agree with the recommendation that a conclusion should be added at the end of the discussion section. We added a brief conclusion section at lines 545-556.

Reviewer 5 Report

Comments and Suggestions for Authors

This study was aimed at describing associations between farm and animal-level variables and AMR in respiratory bacterial isolates from a population of weaned dairy heifers. Despite the fact, that this manuscript presents important scientific insight; the authors are invited to address all the comments of the reviewer.

Areas of concern:

Results

Lines 110: What does hospital milk mean?

Line 125: Put Salmonella in italics

Tables 1 and 2: be consistent in the way the sub-titles are written.

Lines 248-250 and 262-263: how do you explain that the predicted percentage of M. haemolytica isolates not susceptible to florfenicol and macrolides did not follow the same pattern in the hot and cool seasons?

Materials and Methods

Line 519-520: the hot season and cool season were not in the same year: the hot or cool season of the year 2019 could have recorded similar meteorological data. 

Author Response

This study was aimed at describing associations between farm and animal-level variables and AMR in respiratory bacterial isolates from a population of weaned dairy heifers. Despite the fact, that this manuscript presents important scientific insight; the authors are invited to address all the comments of the reviewer.

Areas of concern:

Results

Lines 110: What does hospital milk mean?

Hospital milk is a common term used in the dairy industry for milk that is non-saleable due to disease or treatment in the cow. It is also referred to as “waste milk” We added this additional information at line 110-111.

Line 125: Put Salmonella in italics

We corrected “Salmonella” on lines 100, 101, 125, and in Table 1

Tables 1 and 2: be consistent in the way the sub-titles are written.

We reformatted and changed several things in Tables 1 and 2 to make them consistent. First, we reformatted the width, so they were the same size. Additionally, we updated the bacterial genus and species names to be consistent between the two tables and added the number of isolates and calves total to the sub-title for Table 1 as was done in Table 2.

Lines 248-250 and 262-263: how do you explain that the predicted percentage of M. haemolytica isolates not susceptible to florfenicol and macrolides did not follow the same pattern in the hot and cool seasons?

Thank you for highlighting this as an additional discussion point for the paper. Seasonality has been associated with AMR outcomes in other studies. We have added a section to the discussion on lines 517-526 to discuss associations between seasonality and AMR.

Materials and Methods

Line 519-520: the hot season and cool season were not in the same year: the hot or cool season of the year 2019 could have recorded similar meteorological data. 

We sampled data in two consecutive seasons (we added “consecutive” to line 559). Due to funding availability and study personnel schedules, sampling was commenced in June 2019 and started with the recording of data from the hot season; the following cool season was then in February 2020.

Round 2

Reviewer 2 Report

Comments and Suggestions for Authors

Can be accepted